# Patient level cost of diabetes self-management education programmes: an international evaluation

Gerardine Doyle,[1] Shane O'Donnell,[1] Etáin Quigley,[1] Kate Cullen,[1] Sarah Gibney,[1] Diane Levin-Zamir,[2] Kristin Ganahl,[3] Gabriele Müller,[4] Ingrid Muller,[5] Helle Terkildsen Maindal,[6] Wushou Peter Chang,[7,8] Stephan Van Den Broucke[9]

► Prepublication history and additional material are available. To view these files please visit the journal online (http://dx.doi.org/10.1136/bmjopen-2016-013805).

[1]University College Dublin, , Dublin, , Ireland
[2]Clalit Health Services, , Tel-Aviv, , Israel
[3]Gesundheit Osterreich GmbH, , Vienna, , Austria
[4]Technische Universitat Dresden, , Dresden, , Sachsen, , Germany
[5]Department of Psychology, University of Southampton, , Southampton, , UK
[6]Aarhus University, , Aarhus, , Denmark
[7]School of Public Health, Taipei Medical University, , Taipei, , Taiwan
[8]Department of Occupational Medicine, Shuang Ho and Taipei Medical University Hospital, , Taipei, , Taiwan
[9]Universite catholique de Louvain, Psychological Sciences Research Institute, , Louvain, , Belgium

**Correspondence to**
gerardine.doyle@ucd.ie

## ABSTRACT

**Objectives** The objective of this study was to examine the value of time-driven activity-based costing (TDABC) in understanding the process and costs of delivering diabetes self-management education (DSME) programmes in a multicountry comparative study.

**Setting** Outpatient settings in five European countries (Austria, Denmark, Germany, Ireland, UK) and two countries outside Europe, Taiwan and Israel.

**Participants** Providers of DSME programmes across participating countries (N=16) including healthcare professionals, administrators and patients taking part in DSME programmes.

**Primary and secondary measures** Primary measure: time spent by providers in the delivery of DSME and resources consumed in order to compute programme costs. Secondary measures: self-report measures of behavioural self-management and diabetes disease/health-related outcomes.

**Results** We found significant variation in costs and the processes of how DSME programmes are provided across and within countries. Variations in costs were driven by a combination of price variances, mix of personnel skill and efficiency variances. Higher cost programmes were not found to have achieved better relative outcomes. The findings highlight the value of TDABC in calculating a patient level cost and potential of the methodology to identify process improvements in guiding the optimal allocation of scarce resources in diabetes care, in particular for DSME that is often underfunded.

**Conclusions** This study is the first to measure programme costs using estimates of the actual resources used to educate patients about managing their medical condition and is the first study to map such costs to self-reported behavioural and disease outcomes. The results of this study will inform clinicians, managers and policy makers seeking to enhance the delivery of DSME programmes. The findings highlight the benefits of adopting a TDABC approach to understanding the drivers of the cost of DSME programmes in a multicountry study to reveal opportunities to bend the cost curve for DSME.

## INTRODUCTION

Type 2 diabetes mellitus is one of the major public health threats of the 21st century, currently affecting approximately 59.8 million

---

**Strengths and limitations of this study**

► Time-driven activity-based costing (TDABC) has rarely been applied to care pathways within non-acute settings and as such offers a novel perspective on understanding the costs of providing chronic disease self-management education.

► This is the first multicountry study to compare the costs of diabetes self-management education (DSME) across a number of countries within the European Union and Asia.

► Outcomes of programme participation were measured through self-reported changes, making it difficult to establish if any clinical improvement occurred. Future studies should combine TDABC analysis with clinical outcomes to further assess value in DSME.

---

people within Europe and 415 million worldwide.[1] Further, it has been reported that diabetes medical care accounts for a disproportionate allocation of health service resources across the western world.[1] A recent US study performed an analysis of the spending on personal and public health across 155 conditions across time (1996–2013) and found that spending on diabetes (alongside low back and neck pain) increased the most over this period (US$64.4 billion). Furthermore, the study found that diabetes had the highest healthcare spending in 2013 (US$101.4 billion), a disease attributable to behavioural or metabolic risk factors including diet, obesity, high fasting plasma glucose, tobacco use and low physical activity.[2] Developing the self-care capacity of patients is critical to determining optimal clinical, behavioural and psychosocial outcomes and therefore reducing costs.[3] Diabetes self-management education (DSME) has been shown to improve patient outcomes by reducing the onset and/or advancement of diabetes-related complications; by improving quality of life; strengthening self-efficacy and personal empowerment; assisting with the

development of healthy coping skills; and by reducing diabetes-related distress and depression.[4]

The operation and delivery of DSME varies across the international landscape. They can be either professionally led or peer led. Further, they can be group based, individually based and increasingly IT based. In addition, DSME curricula, duration and delivery may vary substantially, both within and between countries.[5] It is well established that DSME programmes are a low-cost intervention per patient and cost-effective from a payer's perspective. For example, a recent report published by *The Center For Health Law and Policy Innovation* (Harvard Law School) argues that cost savings can be made by public and private insurers in the USA if cost sharing were eliminated and DSME were provided free of charge to patients.[6] However, little research has explored why the costs of running such interventions vary across different healthcare systems and jurisdictions, or why these costs may differ. This study addresses this gap in the prior literature.

Indeed most of the economic analyses have thus far focused on establishing the cost-effectiveness of DSME by comparing the cost of programmes relative to their clinical effectiveness. Such evaluations are usually based on economic modelling, carried out alongside randomised controlled trials, and the findings typically suggest that DSME interventions are cost-effective relative to usual care.[7–13] Despite this, a recent report published by the *Health Information and Quality Authority*[14] in Ireland highlights the large degree of heterogeneity in the methodological approaches used in such economic evaluations. This, in turn, makes results difficult to compare in any meaningful way. In addition, these approaches tend to focus solely on overall cost of running the programmes and neglect to explore potential mechanisms through which DSME programmes could be made more efficient while also maintaining high standards of effectiveness. Furthermore, the majority of studies are based on interventions within a US population, and as such may not be generalisable across differing healthcare, social and cultural contexts.

This study seeks to address these existing gaps in the literature through an economic evaluation of DSME delivery across a number of European Union (EU) and non-EU countries, namely Austria, Denmark, Germany, Ireland, Israel, Taiwan and the UK. The selection of these countries was based on access of the Diabetes Literacy Consortium[*] to local knowledge and networks required to carry out the necessary fieldwork. These countries also represent a diversity of contrasting approaches to the delivery of DSME tailored to each country.[5] The findings are part of a wider study conducted by the Diabetes Literacy Consortium, the overall purpose of which was to examine the (cost)-effectiveness of diabetes education

across Europe, Israel, Taiwan and the USA[†]. The objective of this study is to examine the value of time-driven activity-based costing (TDABC) in understanding the process and costs of delivering DSME in multiple countries and sites (7 countries, 16 sites) and to identify potential process improvements in the delivery of such programmes to reveal opportunities to bend the cost curve for DSME.

## METHOD

A TDABC method was used to map the process of programme delivery and to derive patient level costs.[15 16] TDABC has been developed as a viable costing method for the healthcare sector by Kaplan and Porter[17 18] enabling detailed patient level costs to be computed alongside the identification of possible process improvements resulting in potential cost savings. TDABC is particularly compatible with type 2 diabetes care as the model can be applied to diverse care pathways, particularly chronic disease management. Adopting a TDABC approach in this study therefore gave increased visibility into areas of DSME delivery where process improvements and cost savings could be made, while still maintaining a high quality of patient education. Examples of the application of TDABC have been mostly confined to medical conditions and to acute clinical settings.[18–20] This study seeks to add to this body of knowledge on the costs of care within outpatient environments through identifying the patient level cost of a variety of DSME programmes both *cross*-nationally and *intra*nationally.[21] A primary objective was to provide a robust costing framework within which future studies could include clinical and quality of life outcomes to determine the economic value added to diabetes care through the use of DSME.

The TDABC method involves seven steps[17]: (1) select the medical condition and/or patient population to be examined; (2) define the care value chain; (3) develop process maps of each activity in patient care delivery; identify the resources involved and any supplies used for the patient at each process step; (4) obtain time estimates for each process step; (5) estimate the cost of supplying each patient care resource; (6) estimate the practical capacity of each resource provided and calculate the capacity cost rate (CCR); (7) compute the total costs over each patient's cycle of care. By constructing a sequential activity and process step map and care value chain the researcher can analyse the maps/care pathway for duplication. These areas can then be explored further to establish if changes in the pathway would add value by maintaining/increasing the level of care to the patient while decreasing the economic cost to the overall healthcare system in terms of providing DSME programmes.

---

* The Diabetes Literacy Consortium represents a group of countries funded by the European Commission under the Seventh Framework research programme (Grant Agreement Number: 306186).

† http://www.diabetesliteracy.eu.

Each international study team identified the care value pathway in their country and collected the activity and time data related to the care value pathway, through qualitative semistructured interviews of healthcare providers from each education programme (n=16). These included physicians, nurses, educators and managers. This information was then entered into an aggregated, deidentified database for analysis. All study teams then collected resource and financial data, using a standardised costing worksheet related to the activities, which were then incorporated into the aggregated database for analysis. This methodology was applied to each education programme across each country included in the study. The topic guide was developed in the English language and was then subsequently translated into the local language by the local research teams in each of the participating countries.

All activities associated with the DSME pathway were entered into an aggregated Microsoft Excel database. All activity and time data were collected via the survey instrument, and cost estimates were assigned to these activity variables using financial data provided by the local provider organisations.

DSME programme costs were derived specifically from the cost of performing each activity in the delivery of the programme. All cost data were entered into activity spreadsheets and therefore the data collected did not contain any information relating to identifiable individual service providers. In the resulting database, all cost information was linked to activities and not to individuals. All activity and cost information is reported per DSME programme.

To compare the outcomes of the DSME programmes, a multicentre observational pre-post study design was used involving patients with diabetes enrolled in each of the DSME programmes. Data from the participants were collected at the beginning of the programme and after 3–6 months. The programmes included in the study were existing programmes using five different modes of delivery: individual education in one-on-one sessions, beyond routine treatment provided, group education, self-help groups or a combination of some of the above delivery modes. The content of peer-led and structured DSME programmes was not comparable. Therefore, the two peer-led programmes were excluded from our data analysis.

### Study sample

To be selected for inclusion, programmes had to: (1) target patients with type 2 diabetes; (2) be conducted among the general patient population rather than tailored to the needs of a specific age cohort, needs or gender group; (3) include (but not be limited to) newly diagnosed patients; (4) be stand-alone programmes rather than an add-on to another programme or part of a wider curriculum with (multiple) parallel programmes; (5) admit new patients during the time of the baseline data collection. The study sample size was driven by the number of programmes involved in the

delivery of the specific DSME programmes in each country. Costs were collected for the duration of each programme, which ranged in duration from 1 day to those spanning a 12-month time frame.

### Analytical approach

The TDABC model was used to calculate a cost per programme. Significant variations in programme costs prevailed despite broadly similar programme curricula across countries and programmes. Data collected revealed significant variation in number of education hours across the programmes, number and types of personnel delivering the programmes, practitioner hours and number of participating patients.

Two concepts and measures were drawn upon to develop the TDABC model,[17] the unit cost of supplying capacity and the time it takes to undertake an activity. First, the model was used to calculate the cost of all the resources supplied to each programme. This included personnel, supervision and overheads including rent, equipment, software, and insurance. The total cost was then divided by the actual capacity in order to calculate the cost rate. Second, the CCR was used to assign the programme costs to objects by estimating demand on the resource. Two variables were estimated: the CCR for the programme and the capacity use by each patient. The CCR was calculated by:

$$\text{Capacity Cost Rate} = \frac{\text{Cost of Capacity Supplied}}{\text{Practical Capacity of Resource Supplied}}$$

Practical capacity was used as the denominator in the CCR equation. Estimating the practical capacity required two time estimates which were gathered from Human Resources and other administrative records: the total number of days that each employee actually worked each year; the total number of hours per day that the employee was available for work. Practical capacity was calculated as 80% of this working time.[17] Therefore, 20% was attributed to breaks, training and annual leave. This was applied to all countries to ensure consistency and comparability of the computed programme costs.

In order to calculate the total cost of each DSME programme per patient, the CCRs (including associated support costs) for each resource used were multiplied by the amount of time attributed to each patient. This

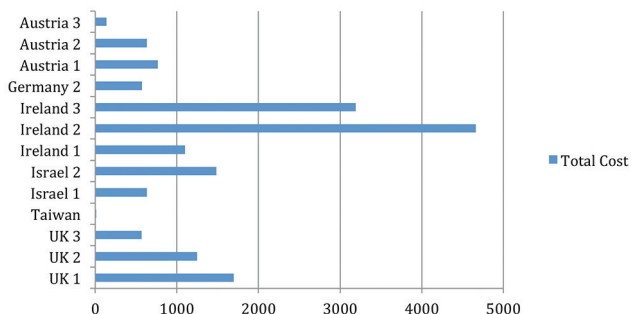

**Figure 1** Cost per programme (salary and overheads) in international dollars.

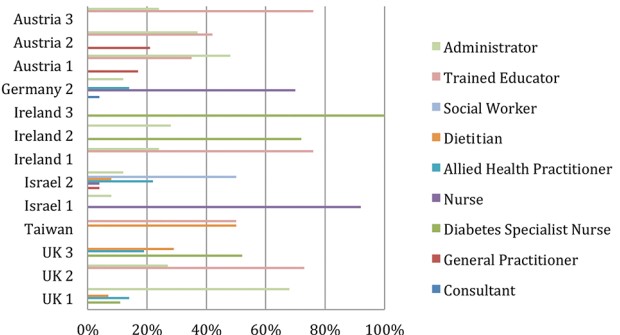

**Figure 2** Percentage of total personnel time used per site.

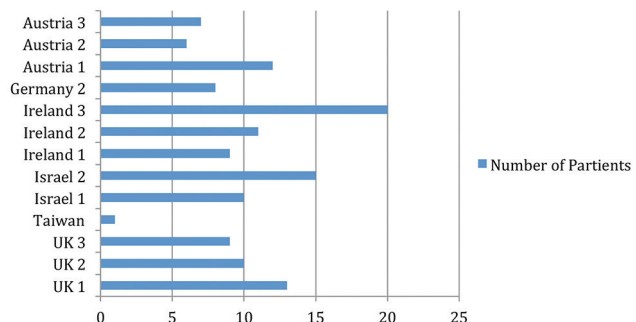

**Figure 4** Number of participants per programme.

calculation was based on the number of patients enrolled at the outset of the programme. The total cost of each programme per patient was the sum of all the costs across all the processes within the DSME programme. The costs were collected in the local currency and then expressed in international dollar to ensure comparability of the costs by using the purchasing power parity conversion factors, to control for different standards of living, different wage levels across countries and for the particular exchange rate.

As suggested by Erhun *et al*, we performed a quantitative investigation of the differences in consumption and pricing of labour resources using cost variance analysis on labour costs. This analysis enabled us to quantitatively discern differences between processes at selected sites. The cost difference can be divided into two effects, a price variance (due to different CCRs of labour resource) and a quantity variance (due to different use of the labour resource across the sites). We performed this variance analysis to understand the differences in consumption and pricing of labour resources and to understand the drivers of cost variation across CCR variances, mix of personnel and efficiency variances[18]

To understand the association between programme cost and health outcomes achieved, we mapped the cost per programme to self-reported patient outcomes. Due to the significant difference in access to clinical data across the participating countries in this study, it was not possible to collect comparable clinical data for each country. Therefore, comparable data were collected to measure outcomes at behavioural and disease/health

outcome levels for existing diabetes self-management programmes. Health outcome data were collected at three levels: individual diabetes self-management disposition, behaviour and disease/health-related outcomes. (The outcome framework employed in this study is summarised in the online supplementary figures, supplementary table 1.)

## RESULTS

Findings highlight that the programmes included in this study provide similar educational content when delivering diabetes education. Further, we found similar changes in self-reported behavioural and disease outcomes across programmes. This suggests that factors other than educational content drive cost variation across programmes and despite reported cost variation, outcomes appear broadly similar. The cost difference between two sites can be analysed into two effects: a price variance due to different CCRs of resource and a quantity variance due to different use of resource:

$$\triangle_{1,2} = \sum_{i=1}^{N_L} q_1^i \times r_1^i - \sum_{i=1}^{N_L} q_2^i \times r_2^i$$

Figure 1 presents the price variance across the sites[‡].

There are a number of factors which were found to influence cost variation. First, programmes differed in duration and hours of practitioner time spent on each programme delivery. This reflects the efficiency variance due to different quantities of total personnel used. For example, figure 2 highlights that the 'Ireland 2' programme uses 78 hours of personnel, whereas 'Austria 3' only uses 5.25 hours of personnel time, yet patient self-reported outcomes are broadly similar. This suggests that total personnel time is a strong cost driver but not an outcome driver. This efficiency variance across two sites is expressed as:

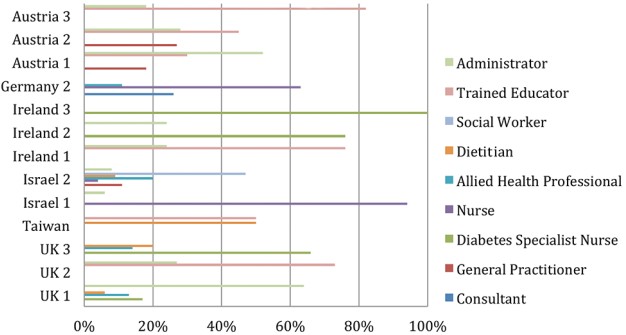

**Figure 3** Weighted average capacity cost rate per site.

‡ For each Figure 1-6 the associated data is included in Supplemental Files attached.

| Programme | Total Cost | Price Variance from Median | Practitioner Hours | Prac Hrs Variance from Median | Weighted Average CCR | Weighted Average CCR Variance from Median | No. of Patients | No. of Patients Variance from Median |
|---|---|---|---|---|---|---|---|---|
| Ire 2 | 4664 | 3,898 | 78 | 55 | 3,825 | 3,022 | 11 | -1 |
| Ire 3 | 3193 | 2,427 | 61 | 38 | 2,515 | 1,712 | 20 | 8 |
| UK 1 | 1697 | 931 | 57 | 34 | 1,560 | 757 | 13 | 1 |
| Israel 2 | 1485 | 719 | 25 | 2 | 482 | -321 | 15 | 3 |
| UK 2 | 1248 | 482 | 13 | -10 | 452 | -351 | 10 | -2 |
| Ire 1 | 1099 | 333 | 19 | -4 | 879 | 76 | 9 | -3 |
| Aus 1 | 766 | 0 | 23 | 0 | 803 | 0 | 12 | 0 |
| Israel 1 | 633 | -133 | 15 | -8 | 346 | -457 | 10 | -2 |
| Aus 2 | 633 | -133 | 19 | -4 | 536 | -267 | 6 | -6 |
| Germany 2 | 573 | -193 | 19 | -4 | 417 | -386 | 8 | -4 |
| UK 3 | 567 | -199 | 12 | -11 | 369 | -434 | 9 | -3 |
| Aus 3 | 136 | -630 | 5 | -18 | 148 | -655 | 7 | -5 |
| Taiwan | 14 | -752 | 5 | -18 | 6 | -797 | 1 | -11 |

**Figure 5** Variance from median programme (Austria 1).

$$= \left( \sum_{i=1}^{N_L} r_2^i \times \frac{q_2^1}{Q_2} \right) \times \left( Q_1 - Q_2 \right)$$

Second, mix of personnel skill used in providing the education is a cost driver. For example, the high salary cost for a consultant physician in Germany and social worker cost in Israel (figure 3) did not produce any significant improvement in patient self-reported outcomes. These findings suggest that personnel skill used is a strong cost driver but does not significantly alter patient self-reported outcomes. When comparing two sites, this mix variance is measured as follows:

$$= \left( \sum_{i=1}^{N_L} \left( \frac{q_1^i}{Q_1} - \frac{q_2^i}{Q_2} \right) \times r_2^i \right) \times Q_1$$

Figure 3 presents the weighted average CCR, the weights representing the percentage of total time used of each personnel type. This highlights both the variety of personnel type used across DSME programmes and countries in addition to the differing percentage of total time used of each personnel type.

Third, the number of patients who attended each programme was a strong per-patient cost driver (figure 4), generally the more patients who attended the programme the lower the per-patient cost.

Taking total cost per programme, the median programme was identified as Austria Programme 1. The key cost drivers identified were then compared with this base programme to explore the behaviour of these variances. Figure 5 summarises this comparison with the base country and reveals that there is a non-linear relationship between the cost of a programme (dependent variable) and each of the key cost drivers (independent variables); practitioner hours used, the weighted average CCR and the number of patients participating on the programmes. In general for practitioner hours, weighted average CCR and patient numbers, as the price variance from the median increases, so too do these independent variables. However, there are some exceptions to this general trend; UK 2 and Ireland 1 where a lower number of practitioner hours are used, Israel 2 and UK 2 where a lower weighted average CCR can be observed, and

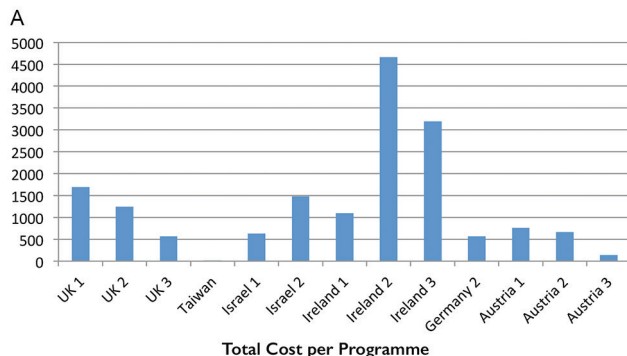

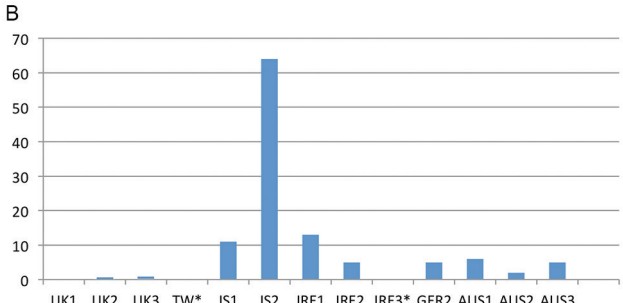

**Figure 6** (**A**) Total cost per programme. (**B**) Change in health outcomes (general diet) following participation across different DSME sites. DSME, diabetes self-management education.

Ireland 2 and UK 2 where there is a lower number of participating patients.

All programmes with a lower cost than the median do have lower practitioner hours, lower weighted average CCR and a lower number of patients, but not proportionately lower. Programme UK 2 appears to be the programme which has a cost higher than the median and yet consistently has a lower number of practitioner hours, lower weighted average CCR and lower patient numbers than the median. This reveals the complexity of the cost behaviours and of the cost variations between the programmes despite offering similar curricula and resulting in similar health outcomes.

There was very little variation in outcomes across each of the programmes, both within and between countries, whatever the mode of delivery, mix of personnel skill used, quantity of total personnel hours, quantity of education hours or quantity of participating patients. For simplicity, figure 6 maps the health outcomes of one particular variable only, general diet, alongside the cost per programme (figure 6A,B). Very modest improvements in general diet were achieved after participation in DMSE and higher cost programmes did not result in better health outcomes. For example, Israel 2 programme recorded the largest change in health outcomes at a low cost in comparison with the most expensive programme, Ireland 2, which resulted in a very small change in health outcomes. Although only general diet is illustrated here, other outcome data show that DSME was only weakly helpful or in some cases had no effect at all on the health outcomes of participants. (Supplemental File Outcomes Framework and Outcomes Data: online supplementary table 2: Self-reported Patient Outcomes)

## DISCUSSION

The data illustrate that DSME programmes are provided at a low cost in every country studied. The data provide evidence that while these costs are low, significant cost variations exist both within and between countries. This is due to a combination of cost variations between the programmes: the CCR, the mix of personnel delivering the education, the different quantities of total personnel used and the number of patients participating in these programmes. This is the first time that such multicountry comparative data have been collected.

The variance analysis performed surrounding costs and outcomes illustrates total personnel hours as a strong cost driver (figure 2). Practitioners such as nurses and diabetes nurse specialists can produce similar outcomes to physicians but at a lower salary and practical capacity cost. This is likely to be a more effective use of resources, particularly in relation to optimising use of personnel at their level of expertise. Further research is needed to explore the most appropriate level of expertise required to deliver the programme for optimal patient health outcomes. For example, instead of having a consultant physician or a Clinical Nurse Specialist delivering the education

programmes, it may be more appropriate to have well-trained experienced nurses or the equivalent performing this role. A pilot study conducted by Kaplan and Porter at The University of Texas Cancer Centre revealed that matching clinical skills to the processes led to a 16% reduction in process time, a 12% decrease in costs for technical staff and a 67% reduction in costs for professional staff.[17] However, clinical outcomes, in addition to behavioural and psychosocial outcomes, are necessary to determine fully whether the educators' level of expertise really influences all DSME health outcomes.[17]

In some countries, the cost of the same programme varied by site. For these programmes, we observed significant variation in administrative hours and this was not associated with the number of participating patients. This finding is similar to that of Muñoz et al who used TDABC in a cost-effective analysis of a red blood cell salvage post total knee arthroscopy in the USA, Switzerland and Austria and suggests that tighter control of administrative costs may reduce what appear to be non-value added (NVA) hours for the patients.[22]

Integrating data on the number of patients participating on each programme (figure 4) with the outcome data suggests that the number of patients in attendance did not impact on patient self-reported outcomes. These findings suggest that there is room for cost savings in DSME regarding the amount of hours of education provided, who provides the education and the number of patients in attendance at each programme, without negatively impacting patient self-reported outcomes.

A number of learnings emerged from this study: first, all programme curricula covered similar topics, this suggests that there is a shared consensus on what information requires dissemination and highlights that variation relates to process delivery rather than curricula; second, the administrative burden on programmes varies greatly and as such is an area of programme development which requires planning and streamlining; third, the skill mix of professionals delivering the programme varies greatly suggesting a lack of empirical knowledge surrounding the most effective educator; fourth, the duration and hours of education varies significantly across sites, again highlighting a lack of consensus in terms of the most efficacious course construct; and finally, such cost variation exists across sites despite the programme content being broadly similar. The granular mapping of the DSME programmes and the derivation of a cost per programme is the first step in generating a better understanding of the DSME arena internationally.

A separate analysis of the self-reported outcome data was conducted by Peer et al analysing the DSME data for all programmes in aggregate.[23] They found that these outcomes were similar irrespective of the education programme or the country (although that the sample size was small and the SD high). When the programmes were analysed in aggregate, a statistically significant improvement was found for six behavioural outcomes (general diet, exercise, medication use, problem areas in diabetes,

foot care and appraisal of diabetes) and three disease/health-related outcomes (body mass index, health-related quality of life and affective well-being). (Please see the Supplemental File Outcomes Framework and Outcomes Data, online supplementary table 3, and related note explaining the precise scales used.)

The costing and provision of DSME is at an early stage of development globally with limited empirical knowledge of the most efficient and effective mode of delivering DSME. Thus, this study has gone some way to remedying this problem where it has outlined a bottom-up/patient level cost using estimates of the actual resource costs used to educate patients through self-management programmes and therefore a more accurate cost than heretofore of providing various education programmes. Thus, it has provided a first layer of information, which in the future will be required to establish whether this model of care/intervention can add value to the healthcare system once clinical effectiveness outcomes have been determined for each programme. Storfjell et al show that the application of TDABC in the context of nursing care can facilitate the identification and elimination of NVA time and related the increase in time spent on psychosocial intervention, support and patient education.[24] However, there is a long way to go, where clinical and quality of life outcomes are required to measure the effectiveness of DSME programmes before a thorough understanding of their added value to patients can be estimated. The methods and results of the current study will inform future research to achieve a better understanding of the added value derived from providing DSME interventions. We suggest that future studies include a rigorous collection of clinical outcomes pre and post DSME.

## LIMITATIONS

The TDABC method is a relatively new method in terms of healthcare costing, and to the best of the authors' knowledge has yet to be applied to investigate the costs of a health education intervention. As a result, there were limited guidelines surrounding the collection of activity and process step data in non-acute settings, and thus it was necessary for the research team to develop such a protocol that was fit for purpose across different international study sites. In practice, many participants were unfamiliar with the costing and activity terminology and the level of detail required on all forms of activity, for TDABC. We observed that participants appeared to provide less detail on administrative and programme preparation activity compared with education activity. This detailed information would have provided greater insight into the reasons why administrative costs were found to be so high in some countries while not in others.

In addition, some of the local research teams also experienced difficulties in collecting the required financial data. For example, in Belgium, the staff involved in the delivery of DSME programmes taking part in this study were unable to disclose personal salary information, which was not otherwise available from a public source, as in other countries. This related to data protection legislation (enacted 1992, subsequently amended 1998, 2003), together with the fact that there is no professional category of diabetes educator in Belgium. For these reasons, the Belgian data had to be excluded from this particular study.

The study is also limited by a lack of available clinical outcome data from each of the education programmes. While important self-reported health and psychosocial outcome data were collected in each country, it was not possible to determine the clinical effectiveness of these DSME programmes in terms of glycaemic control due to the absence of any clinical measures. As Kaplan and Porter point out,[17] value in healthcare can only be determined when there is visibility into both costs and *clinical* outcomes. Furthermore, the reliability of self-reported outcomes data was undermined by small sample sizes in each country. Second, self-reported measures of health behaviour are susceptible to social desirability bias, and response styles can vary by culture and setting.[25 26] Nonetheless, the similarity in outcomes across each of the sites regardless of the amount of money invested in each programme raises questions surrounding the value being achieved per euro/dollar spent.

The peer-led programmes found in Denmark and Germany were excluded from the analysis. However, they were provided at the lowest cost of international $0.15 and $0.74 per patient per hour of education, respectively. When self-assessed outcome data were measured for each programme, the outcomes were similar for peer-led and specialist-led programmes. We suggest that further research is needed surrounding peer-led education and measurement of associated clinical health outcomes.

## CONCLUSION

This paper has demonstrated the variances in the cost of delivering different types of diabetes education programmes, both within and across countries in the EU and Asia. Developing cost-effective lifestyle interventions to improve the diabetes knowledge and self-management skills and quality of life for patients may be an important step in preventing the onset of complications associated with type 2 diabetes. The imperative to do so from an economic perspective cannot be underestimated when consideration is given to the implications for healthcare systems associated with the treatment of diabetes-related morbidities such as active foot disease, chronic kidney disease, retinopathy and myocardial infarction.[27]

This study offers the first application of a TDABC approach to evaluate the cost of delivering DSME programmes and as a means of comparing the costs of running a healthcare intervention cross-nationally. It contributes to the extant literature by highlighting and describing the vast combinations and permutations of

delivery of DSME curricula, practitioner hours, hours of education, mix of educators, numbers of attendees and how these variations lead to substantial cost differences. Our variance analysis revealed that the key drivers of cost variation arose from differing weighted average CCRs representing the percentage of total time used of each personnel type, the mix of personnel delivering the education and the number of patients participating in these programmes. In the process, we identified how there could be potentially unnecessary process steps that, if eliminated, could lead to cost savings in the delivery of DSME programmes, including vast differences in administration time, and exploring the mix of personnel skill alongside the total personnel time used.

While it is already established that diabetes education is a low-cost intervention and is cost-effective, given the sheer numbers of education programmes that need to be made available to meet the demands resulting from increasing levels of diabetes worldwide, even small process improvements could lead to overall cost savings for healthcare providers. Future studies focusing on the cost-effectiveness of healthcare interventions may consider adopting TDABC principles and variance analysis as a means of identifying efficiencies in other chronic disease education programmes.

The study has highlighted the strengths of TDABC as a method of bottom-up costing in outpatient care and recommends using this method in future studies so as to allow for a comprehensive literature to develop in the area, enabling comparative studies to be performed. By developing such literature, a comprehensive understanding of the cost of patient education programmes can be developed and compared cross-nationally and across time. Healthcare practitioners and educators who wish to convince policy makers and health insurers to reimburse the cost of DSME delivery can adopt a TDABC approach in order to demonstrate that such programmes are run efficiently and effectively especially when combined with measures of consequent clinical health outcomes to represent value for money.

**Correction notice** This paper has been amended since it was published Online First. Owing to a scripting error, some of the publisher names in the references were replaced with 'BMJ Publishing Group'. This only affected the full text version, not the PDF. We have since corrected these errors and the correct publishers have been inserted into the references.

**Acknowledgements** The authors would like to thank the partners of the Diabetes Literacy Consortium for contributing to data collection with participants and providers of DSME programmes in their respective countries. Université Catholique de Louvain, Belgium: Stephan Van den Broucke, Gerard Van der Zanden, Marie Housiaux, Louise Schinckus. Technische Universität Dresden, Medical Faculty, Germany: Peter Schwarz, Gabriele Mueller, Henna Riemenschneider, Sarama Saha. University College Dublin, Ireland: Gerardine Doyle, Shane O'Donnell, Etain Quigley, Kate Cullen, Sarah Gibney. Gesundheit Österreich GmbH, Austria: Jürgen Pelikan, Florian Röthlin, Kristin Ganahl, Sandra Peer. Maastricht University, Department of International Health, The Netherlands: Helmut Brand, Kristine Sörensen, Timo Clemens, Marjo Campmans. University of Southampton, UK: Lucy Yardley, Ali Rowsell, Ingrid Muller, Victoria Hayter. Clalit Health Services, Israel: Diane Levin-Zamir, Ziv Har-Gil. University of California at San Francisco, USA: Dean Schillinger, Courtney Lyles, Lina Tieu. Taipei Medical University, Taiwan: Peter Chang, Candy Kuo, Alice Lin, Duong Van Tuyen, Becky Sun. Aarhus University,

Denmark: Helle Terkildsen Maindal, Jill Rowlands, Ulrik Martensen. We also thank all the medical and educational services from the eight participating countries who contributed to the data collection.

**Contributors** GD initially proposed the study. GD and SG specified the methodology. GD and KC carried out the cost analysis. All authors contributed to the protocol design, data collection and analysis plan. EQ and SOD wrote the initial manuscript, and all authors contributed to improving the manuscript. All authors approved the final manuscript.

**Funding** This study is part of the Diabetes Literacy project supported by grant FP7-Health-2012-Innovation 1/306186 of the European Commission.

**Competing interests** None declared.

**Patient consent** Obtained.

**Ethics approval** All methods were approved by the SVUH Group Research and Ethics Committee, by the Research Ethics Committee of the Office of Research Ethics, University College Dublin, and by the relevant local ethics committee in each jurisdiction and each study site where the study was carried out.

**Provenance and peer review** Not commissioned; externally peer reviewed.

**Data sharing statement** The Excel spreadsheets showing how the individual activity costs were aggregated are available should they be requested.

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
