## [Reviewer comments · BMJ Open]

ARTICLE DETAILS

TITLE (PROVISIONAL)	Patient Level Cost of Diabetes Self-Management Education Programmes: An International Evaluation
AUTHORS	Doyle, Gerardine; O'Donnell, Shane; Quigley, Etáin; Cullen, Kate; Gibney, Sarah; Levin-Zamir, D; Ganahl, Kristin; Mueller, Gabriele; Muller, Ingrid; Maindal, Helle; Chang, Wushou; Van Den Broucke, Stephan

VERSION 1 - REVIEW

REVIEWER	Robert S. Kaplan Harvard Business School USA
REVIEW RETURNED	01-Sep-2016

GENERAL COMMENTS	The paper has several important strengths. It addresses the management of an important medical condition, diabetes, whose incidence is rapidly increasing around the world. It is an ambitious multi-country comparative study that compares the costs of 16 disease management educational programs in 7 different countries in Europe and Asia. It is the first to measure program costs using estimates of the costs of the actual resource costs used to educate patients about how to treat their medical condition through self-management programs. The current draft, however, has several limitations and shortcomings that could be overcome in a revised paper. I have identified the following improvement opportunities: 1. A significant and inherent limitation is that the study did not collect any outcomes data from patients who were trained in the DSME programs. Cost variation could occur because the more expensive programs are delivering better education and their patients have far lower treatment and complication costs, which would dwarf the higher incremental costs of the more expensive programs. Because of the lack of outcomes data, the current paper must excel at the methodology used to compare cost variations across the 16 international programs; the current draft falls somewhat short of this standard.2. Accepting the lack of comparative outcomes data, are the DSME programs themselves standard across the sites? Some of the documented variation could be caused by delivering very different programs. That different educational programs cost different amounts would hardly be a surprising or publishable finding. The methodology of the study would seem to require that the content of the programs are comparable, so that the source of cost variation is due to alternative ways of delivering the same content. But this is not made clear in the paper.
---

	3. The paper features, as a major dependent variable, the cost per patient hour. It is not clear why this is an interesting metric. Assuming that the contents of the program are the same, then the relevant dependent variable would be the education cost per patient. If the authors believe that cost per patient hour is highly relevant, they should provide an explanation for why they believe this to be true. 4. I recommend providing graphic summaries of the cost variation documented in Table 2. One chart would be clearer to quickly communicate the cost variation across the 16 programs; perhaps graph the costs of the 16 observations (labeled) on a vertical (or horizontal) scale. 5. Much of the text in lines 249-328 repeats information already in Table 2. The authors should provide commentary only when it adds to the information already existing in Table 2. 6. Line 335: "costs differ as a result of these variances in approaches." That costs vary across programs is mildly interesting but not surprising. Can the authors identify specifically what factors cause costs to be either higher or lower? Such an analysis would inform the reader to important correlations across the 16 sets of observations. 7. Big Recommendation: The authors should do a variance analysis to control for the different employee costs per hour of the clinicians and administrators at the different sites. Some to much of the cost variation is due to the different wage levels across countries and to the particular exchange rate used to convert prices in different countries to a base level. If all employee types are costed at the same rate, than (say the average across the 16 sites, or the actual rates in a base country (e.g., Ireland) the remaining variation across the sites can be attributed to differences in productivity, efficiency, and mix of personnel types used in the programs. The use of variance analysis to control for inter-site differences in wage levels and currencies was published last year in BMJ Open: "Time-driven activity-based costing of multi-vessel coronary artery bypass grafting across national boundaries to identify improvement opportunities: study protocol," BMJ Open. 2015. 5:8 e008765 doi:10.1136/bmjopen-2015-008765. F Erhun, B Mistry, T Platchek, A. Milstein, V G Narayanan, R S Kaplan. 8. The productivity/efficiency variance, calculated in recommendation 7. can be further decomposed into a pure quantity variance and a mix variance (using lower skilled personnel to perform the same function). 9. How much of the cost per patient is influenced by the number of patients in each program? This would seem to be another driver of variation in the cost per patient in each program.
--	---

REVIEWER	Adriana Coppola, PhD Clinical Institute Beato Matteo, San Donato Group, Vigevano, Italy
REVIEW RETURNED	31-Oct-2016

GENERAL COMMENTS	The study by Doyle and colleagues aimed at analysing the real costs of delivering DSME programs in seven countries by using the TD-ABC method which was used in previous studies to evaluate costs in other clinical conditions. The present investigation provides a very original and interesting "photograph" of the costs linked to diabetes
--

	patient education, but it fails to give any additional information. Indeed, I believe that it is not possible to state that TD-ABC gives benefits to review and evaluate health care services in diabetes; to correctly do a cost-effective analysis, the impact of costs on clinical outcomes should be also evaluated. In addition, a comparison between methods should be made. At last the simple analysis of the costs cannot permit any discussion about how to plan future DSME programmes or strategies, if clinical outcomes are not analysed. This implies that the discussion should be fully re-written and shortened at least by 30%. I think that the correct information obtained from the present study has been adequately summarized by the same authors in the "article summmary". No additional information was given. At last, I recommend to eliminate the last sentence from the abstract. Additional comment. Aim should be correctly written at the end of the introduction by using the style adopted in the abstract. It may be useful to avoid a series of questions
--	---

VERSION 1 – AUTHOR RESPONSE

Reviewer: 1

Reviewer Name: Robert S. Kaplan

Institution and Country: Harvard Business School, USA

Competing Interests: None declared

The paper has several important strengths. It addresses the management of an important medical condition, diabetes, whose incidence is rapidly increasing around the world. It is an ambitious multi-country comparative study that compares the costs of 16 disease management educational programs in 7 different countries in Europe and Asia. It is the first to measure program costs using estimates of the costs of the actual resource costs used to educate patients about how to treat their medical condition through self-management programs.

The current draft, however, has several limitations and shortcomings that could be overcome in a revised paper. I have identified the following improvement opportunities:

Comment 1

A significant and inherent limitation is that the study did not collect any outcomes data from patients who were trained in the DSME programs. Cost variation could occur because the more expensive programs are delivering better education and their patients have far lower treatment and complication costs, which would dwarf the higher incremental costs of the more expensive programs. Because of the lack of outcomes data, the current paper must excel at the methodology used to compare cost variations across the 16 international programs; the current draft falls somewhat short of this standard.

Response 1

Thank you for this important comment. We have now addressed the two issues of concern by including the outcomes data we did collect and by improving our analysis of the cost drivers and reasons for cost variations using variance analysis as described in Erhun et al., (2015) BMJ Open paper.

Due to the significant variance in access to clinical data across the participating countries in this study, it was not possible to collect comparable clinical data for each country. Therefore comparable data was collected to measure outcomes at behavioral and disease/health outcome levels for existing

diabetes self-management programs. (The outcome framework employed in this study is now included in the Supplemental Files.)

We found that these outcomes were similar irrespective of the education program or the country (albeit that the sample size was small and the standard deviation high). Following attrition between the base line and follow-up 3-6 months of 25%, alongside pre-defined exclusion criteria, the overall sample size for outcome data analysis was 366 (58%) patients across the nine countries.

Across each programme, a statistically significant improvement was found for six behavioral outcomes (general diet, exercise, medication use, problem areas in diabetes, foot care and appraisal of diabetes) and three disease/health outcomes (BMI, health related quality of life and affective well-being).

The biggest increase was found for foot care: patients who participated in a DSME program took care of their feet more than half a day more during an average week than before the program. Participants also showed a more positive appraisal strategy and better problem solving. The proportion of participants who were physically active and who follow the prescribed medication seven days per week was also greater after the program than before. With regard to the disease/health outcomes, the average health related quality of life and affective well-being increased, and mental comorbidity decreased.

While we observed very little difference in the behavioral and disease/health outcomes of between the various education programs, we did find large cost deviations across the programs and within countries. We have now summarised these findings in the revised paper and have presented the outcome data in Supplemental files.

The cost variation therefore does not appear to be explained by the more expensive programs delivering better education. To understand the source of cost variation, variance analysis is now presented in the Results section of the paper (Section 3).

Comment 2

Accepting the lack of comparative outcomes data, are the DSME programs themselves standard across the sites? Some of the documented variation could be caused by delivering very different programs. That different educational programs cost different amounts would hardly be a surprising or publishable finding. The methodology of the study would seem to require that the content of the programs are comparable, so that the source of cost variation is due to alternative ways of delivering the same content. But this is not made clear in the paper.

Response 2

We have now stated clearly in the paper that the content of the programs are broadly similar and therefore comparable, and that the source of cost variation is due to alternative ways of delivering the same content (lines: 273-277). We have now clarified that the educational content of the DSME programs is standard across countries and sites and that the variation therefore is not caused by delivering very different programs.

In this regard we have now excluded from our analyses three programs which do not offer the same content, the two peer led programs (Denmark and Germany 3) and Germany 1 which only addressed nutritional topics.

The variance analysis we have now performed in Section 3 demonstrates the cost drivers to explain the cost variation across programs. These include the quantity variance and the capacity cost rate variance to quantitatively discern differences between the processes of education delivery across the

13 sites. We have further analysed the quantity variance into the mix variance where the programs use a different mix of personnel and the efficiency variance to reflect the cost differences arising from the total quantity of labour used.

Comment 3

The paper features, as a major dependent variable, the cost per patient hour. It is not clear why this is an interesting metric. Assuming that the contents of the program are the same, then the relevant dependent variable would be the education cost per patient. If the authors believe that cost per patient hour is highly relevant, they should provide an explanation for why they believe this to be true.

Response 3

We have now clarified that the education cost per patient is indeed a major dependent variable because the education content is broadly similar across the sites. Our original rationale for computing the cost per patient per education hour had been to control for the significant variation in education hours across the programmes. This of course may be addressed through variance analysis as suggested under Comment 7 below. We have now performed this variance analysis to provide a deeper understanding of the reasons for the cost variations arising. Please find this analysis in Section 3 of the revised paper, which has now been completely rewritten.

Comment 4

I recommend providing graphic summaries of the cost variation documented in Table 2. One chart would be clearer to quickly communicate the cost variation across the 16 programs; perhaps graph the costs of the 16 observations (labeled) on a vertical (or horizontal) scale.

Response 4

Thank you, this is an excellent suggestion. We have now removed Tables 1 and 2 from the paper and replaced them with graphic summaries of the cost variations. Figure 1 shows the comparative education cost program (expressed in international dollars). Figures 2 and 3 further analyse the cost difference into the efficiency and the mix variances. We hope that these graphs provide the reader with a clearer summary of the findings.

Comment 5

Much of the text in lines 249-328 repeats information already in Table 2. The authors should provide commentary only when it adds to the information already existing in Table 2.

Response 5

We thoroughly agree that this text is unnecessary. We have now deleted all the text in lines 249-328 and completely rewritten this section of the paper.

The findings are now summarised in a number of graphs providing only necessary commentary. The supporting data is included in Supplemental Files. All discussion of the findings is now contained in Section 4, the discussion section.

Comment 6

Line 335: "costs differ as a result of these variances in approaches." That costs vary across programs is mildly interesting but not surprising. Can the authors identify specifically what factors cause costs to be either higher or lower? Such an analysis would inform the reader to important correlations across the 16 sets of observations.

Response 6

The variance analysis now performed in Section 3 has enabled us to identify specifically what

factors/processes cause costs to vary between the now 13 observations. These include the quantity of total labour hours, the differing capacity cost rate, the mix of personnel delivering the programs and the number of patients participating on each program.

Comment 7

Big Recommendation: The authors should do a variance analysis to control for the different employee costs per hour of the clinicians and administrators at the different sites. So much of the cost variation is due to the different wage levels across countries and to the particular exchange rate used to convert prices in different countries to a base level. If all employee types are costed at the same rate, than (say the average across the 16 sites, or the actual rates in a base country (e.g., Ireland) the remaining variation across the sites can be attributed to differences in productivity, efficiency, and mix of personnel types used in the programs. The use of variance analysis to control for inter-site differences in wage levels and currencies was published last year in BMJ Open:

“Time-driven activity-based costing of multi-vessel coronary artery bypass grafting across national boundaries to identify improvement opportunities: study protocol,” BMJ Open. 2015. 5:8 e008765 doi:10.1136/bmjopen-2015-008765. F Erhun, B Mistry, T Platchek, A. Milstein, V G Narayanan, R S Kaplan.

Response 7

Thank you for this recommendation, which has improved the quantitative analysis of the data collected.

By converting all currencies to the international dollar, using the Purchasing Power Parity conversion factors, we were seeking to control for different standards of living, different wage levels across countries and for the particular exchange rate used to convert prices in different countries. We have now expressed all education cost data in international dollars throughout the paper.

In addition we have chosen the median site across the 16 sites as the benchmark site and have performed variance analysis to understand how the variation across the sites may be attributed to differences in price, efficiency, and mix of personnel types used in the programs.

Comment 8

The productivity/efficiency variance, calculated in recommendation 7 can be further decomposed into a pure quantity variance and a mix variance (using lower skilled personnel to perform the same function).

Response 8

The variance analysis performed in Section 3 now addresses the pure quantity variance and its decomposition into the mix and efficiency variances. The results of this variance analysis have now been included in the discussion section of the paper (Section 4).

Comment 9

How much of the cost per patient is influenced by the number of patients in each program? This would seem to be another driver of variation in the cost per patient in each program.

Response 9

We agree that the number of patients in each programs may be a driver of cost variation. This is summarised in Figure 4.

This reflects our original rationale for computing an education cost per patient per program. On reflection this is more appropriately presented using variance analysis which has been included in the

revised paper.

Thank you for your valuable comments and suggestions, which we have now reflected in the revised paper. We hope that the greater attention to quantitative analysis and the inclusion of graphic summaries have significantly improved the paper.

Reviewer: 2

Reviewer Name: Adriana Coppola, PhD

Institution and Country: Clinical Institute Beato Matteo, San Donato Group, Vigevano, Italy

Competing Interests: None declared

Comment 1

The study by Doyle and colleagues aimed at analysing the real costs of delivering DSME programs in seven countries by using the TD-ABC method which was used in previous studies to evaluate costs in other clinical conditions. The present investigation provides a very original and interesting "photograph" of the costs linked to diabetes patient education, but it fails to give any additional information. Indeed, I believe that it is not possible to state that TD-ABC gives benefits to review and evaluate health care services in diabetes; to correctly do a cost-effective analysis, the impact of costs on clinical outcomes should be also evaluated.

Response 1

Thank you for this comment. Perhaps we did not state clearly at the outset that this paper is a costing paper and a cost analysis paper and is not a cost-effectiveness paper. Rather it is an example of the application of TDABC in a novel way to Diabetes Self-Management Education programs instead of a medical condition. It is an ambitious multi-country comparative study that compares the costs of 16 disease management educational programs in 7 different countries within Europe and Asia. It is the first study to measure program costs using estimates of the costs of the actual resource costs used to educate patients about how to treat their medical condition through self-management programs.

The purpose of this paper was to examine the value of Time Driven Activity Based Costing (TDABC) in understanding the process and costs of delivering Diabetes Self-Management Education programs (DSME). We seek to understand how the cost variation across the sites may be attributed to differences in productivity, efficiency, and mix of personnel types used in the programs and to identify potential process improvements in the delivery of such programs.

This study is a rigorous application of TDABC to generate first time data about the costs of DSME programs and is the first study to attempt to link such costs to health outcome data, albeit self-reported behavioral and disease/health outcome data for existing diabetes self-management programs rather than clinical outcome data. Clinical health outcomes were the ideal data to collect, however not all countries participating in this comparative study had access to such data. This is a clear limitation of large comparative studies.

We have now included the health outcome data in the paper and we hope that our revised paper more clearly states the contribution of this study to prior work.

Comment 2

In addition, a comparison between methods should be made.

Response 2

There is an abundance of literature surrounding the benefits of performing bottom up costing approaches such a TDABC rather than a top down traditional allocation of costs. Given the word limit

of 4000 words we did not believe it was appropriate to rehearse these arguments in detail. TDABC as a suitable method for deriving a patient level cost is discussed in the method section of the paper (Section 2) and a comparison between top down and bottom up costing methods is briefly discussed within the Discussion section (Section 4).

We set out to adopt the TDABC approach to costing DSME programs, which has not been done in prior studies and having done so to compare the costs of delivering such programs in different countries and different sites (16 sites) and consequently to identify and understand the drivers of cost variation. We have now performed more detailed variance analysis to understand the reasons for such cost variation.

Comment 3

At last the simple analysis of the costs cannot permit any discussion about how to plan future DSME programmes or strategies, if clinical outcomes are not analysed.

Response 3

We have now performed a more in-depth quantitative investigation of the differences in consumption and pricing of labour resources between the program sites using variance analysis on labour costs. Please see Section 3 of the revised paper.

By mapping the process of education delivery for each of the 16 programs using the TDABC method, we believe the data gathered and analysed would be beneficial to the future planning of DSME programmes and strategies, for example surrounding the skill mix of personnel delivering the education, the total quantity of personnel hours and the monitoring and control of administration hours. We agree that we did not highlight this sufficiently in the earlier paper.

Furthermore we have now included the self-reported health outcome data in our analysis. This part of the study revealed that while a statistically significant improvement was found for six behavioral outcomes (general diet, exercise, medication use, problem areas in diabetes, foot care and appraisal of diabetes) and three disease/health outcomes (BMI, health related quality of life and affective well-being) there was little variation in these outcomes across the 16 programs. This suggests that cost variation is not explained by better delivery of education i.e. that the more expensive programs do not yield better health outcomes for patients.

Comment 4

This implies that the discussion should be fully re-written and shortened at least by 30%.

Response 4

Thank you, both the Results and Discussion sections have been shortened and replaced with a more in-depth discussion of the variance analysis performed and the contribution of this study as a multi-country comparative study that compares the costs of 16 disease self-management education programs in 7 different countries within Europe and Asia. It is the first study to measure program costs using estimates of the costs of the actual resource costs used to educate patients about how to treat their medical condition through self-management programs.

Comment 5

I think that the correct information obtained from the present study has been adequately summarized by the same authors in the "article summary". No additional information was given.

Response 5

We hope that the paper is much improved following the more concise graphical presentation of our findings together with the in-depth quantitative cost variance analysis to include the capacity cost rate

variance and quantity variance (mix and efficiency variances). On reflection the previous version of the paper did not properly reflect the granular data collected and was discussed more from a qualitative rather than a quantitative perspective. We hope that the greater attention to quantitative analysis has significantly improved the paper.

Comment 6

At last, I recommend to eliminate the last sentence from the abstract.

Response 6

Thank you, we have amended the abstract as suggested and in light of our improved cost variance analysis.

Comment 7

Additional comment.

Aim should be correctly written at the end of the introduction by using the style adopted in the abstract. It may be useful to avoid a series of questions.

Response 7

Thank you, we have now restated the aim of the study at the end of the Introduction section using the same style as that set out in the abstract.

Thank you for your valuable comments and suggestions. We hope that our paper is now much improved.

VERSION 2 – REVIEW

REVIEWER	Robert S. Kaplan Harvard Business School U.S.A.
REVIEW RETURNED	05-Jan-2017

GENERAL COMMENTS	This revision is a major improvement over the original one I reviewed. The author team responded very well and constructively to the concerns and suggestions of the two reviewers. I still have a few suggestions or questions that I describe below. I would like to see them addressed before granting a final acceptance. Line 100: A very recent reference on the high cost of diabetes care in the U.S. can be found at http://jamanetwork.com/journals/jama/fullarticle/2594716 . Line 148: Prefer to replace "with the health care sector" with "for the health care sector". Line 251: Add "variance" after "price". Line 309: Total column in Figure 3 data does not make any sense. The only summary statistic that would be meaningful would be a weighted average of the CCRs at each site, with the weights representing the percentage of total time used of each personnel type (weights should sum to 100%). Also, the CCR for Social Workers at Israel 2 seems much too high; how could it be 2x the cost of a physician. Also, at this site, why do GPs, AHPss, Dieticians, and Nurses have the same CCR to four significant digits? This is unlikely to be correct. Please check these
---

	underlying data for accuracy. Lines 322-323: Just looking at the bar chart on Figure 5, we can see high variability in cost across programs but the "nonlinear relationship" is not obvious. What is the independent variable for which the dependent variable has a nonlinear relationship. Line 333: The graphical representation of Figure 6 has two problems. First, choose a scale for outcomes that at least has the potential to see the variation in outcomes (the green bars) across the 11 sites. At present, all we see are green dots, and nothing happening on the vertical scale. Second, how are the diverse outcome metrics in "Figure 6 Data" aggregated into a single outcome metric score? For the supporting data for Figure 6 Data, we need an explanation about the scale used by patients in reporting the PROs. The analysis claims that the outcomes are similar across programs, but does not address that the outcomes are also very similar to 0 (no impact from DMSE). Is the preponderance of positive signs used as the evidence that the DMSEs are weakly helpful, even though with the small sample sizes, virtually none of the changes is more than 1 SD away from 0. I don't understand the statement on lines 394-5 about statistically significant outcomes; only a very few of the reported means is more than 2x the standard deviation (what I assume SD stands for). I also don't see the data, in "Figure 6 Data" on the PROs of BMI, health related quality of life and affective well-being (lines 396-7). Lines 430-433: It is unfortunate, but not fatal, that Belgium refused to supply confidential salary data. The Belgian study group could use data from public salary surveys of compensation of each type of employee. This would be approximately accurate, and good enough for the analysis in this paper, while not requiring that sensitive salary data be disclosed. I would encourage the Belgium group to access and apply public compensation survey data so they can rejoin the DMSE project team and paper.
--	---

REVIEWER	Adriana Coppola Clinical Institute Beato Matteo, Vigevano, Italy
REVIEW RETURNED	10-Jan-2017

GENERAL COMMENTS	My concerns have been adequately addressed
--

VERSION 2 – AUTHOR RESPONSE

Reviewer: 1

Reviewer Name: Robert S. Kaplan

Institution and Country: Harvard Business School, USA

Competing Interests: None declared

Reviewer Comments

This revision is a major improvement over the original one I reviewed. The author team responded very well and constructively to the concerns and suggestions of the two reviewers. I still have a few suggestions or questions that I describe below. I would like to see them addressed before granting a final acceptance.

Comment 1

Line 100: A very recent reference on the high cost of diabetes care in the U.S. can be found at <http://jamanetwork.com/journals/jama/fullarticle/2594716> .

Response 1

Thank you for alerting us to this very important study. We have discussed the findings in lines 100-105 and referenced the paper accordingly.

Comment 2

Line 148: Prefer to replace "with the health care sector" with "for the health care sector".

Response 2

Thank you, this has now been amended (now line 154).

Comment 3

Line 251: Add "variance" after "price".

Response 3

Thank you, this has now been added (now line 259).

Comment 4

Line 309: Total column in Figure 3 data does not make any sense. The only summary statistic that would be meaningful would be a weighted average of the CCRs at each site, with the weights representing the percentage of total time used of each personnel type (weights should sum to 100%).

Response 4

The Total Column in Supplemental File Data for Figure 3 has now been updated to present the weighted average of the CCR's for each personnel type at each site. For this purpose the Supplemental File Data for Figure 2 has also been amended to show the % of time for each personnel type per site. The Figure 3 graph has likewise been updated to present the weighted average of the CCRs at each site, with the weights representing the percentage of total time used of each personnel type. Lines 311-314 of the manuscript have been amended accordingly.

Comment 5

Also, the CCR for Social Workers at Israel 2 seems much too high; how could it be 2x the cost of a physician. Also, at this site, why do GPs, AHPs, Dieticians, and Nurses have the same CCR to four significant digits? This is unlikely to be correct. Please check these underlying data for accuracy.

Response 5

Thank you, we do agree with the reviewer. There had been many changes to the cost data supplied by the Israel team during the study. We have now contacted our partners in Israel to confirm that the data originally supplied for the costing of their DSME offerings are the correct costs. All tables and graphs have been updated accordingly.

Please note that the correct salary figures show that some health care professionals do indeed have

some very similar salaries in Israel. For example, the salary per hour for a Dietician, Nurse, Health Promoter and a Social Worker are very similar (ILS 118, ILS 117, ILS 115 and ILS 105 respectively). Physiotherapist is a little higher at ILS 148 and a GP at ILS 308 per hour.

Comment 6

Lines 322-323: Just looking at the bar chart on Figure 5, we can see high variability in cost across programs but the "nonlinear relationship" is not obvious. What is the independent variable for which the dependent variable has a nonlinear relationship.

Response 6

The Supplemental File for Figure 5 Data highlights variance between the dependent variable (cost) and the independent variables (practitioner hours, weighted average capacity cost rate and number of patients).

The accompanying Figure 5 bar chart has been amended. However due to the high values for the dependent variable, the variances from the median are not very obvious on the bar chart. We have concluded that perhaps the table of data from the Supplemental File for Figure 5 Data best illustrates this non-linear relationship rather than by the graph. We have now included the table of data in the paper as Figure 5 and show the graph in the Supplemental File for your review.

The analysis of the relationship between the dependent and independent variables has now been included in the manuscript, lines 325 – 336.

Comment 7

Line 333: The graphical representation of Figure 6 has two problems. First, choose a scale for outcomes that at least has the potential to see the variation in outcomes (the green bars) across the 11 sites. At present, all we see are green dots, and nothing happening on the vertical scale.

Response 7

Our apologies, it was within the Supplemental File Outcomes Framework and Outcomes Data that we had included an explanation by way of footnote as follows:

‘For Figure 6, General Diet was taken as one example of the health outcome data achieved when mapped with cost per programme. To include each health outcome would make Figure 6 too complex and the main finding that health outcomes were similar across all programmes would not be clear to the reader.’

We now realise that we should also have explained this within the manuscript. We have now clarified this in lines 344-353.

Due to the variety of scales used in the study to measure health outcomes, it was not possible to choose a single scale for all outcomes. (Please see Supplemental File Outcomes Framework and Outcomes Data, footnotes to Tables 1-3 for the precise scales used.) This explains why we chose one health outcome, general diet, to represent the pattern we observed, that health outcomes were similar across all programmes.

We have amended the visual presentation of the programme total cost and the illustrative General Diet health outcome by showing two bar charts alongside each other (Figure 6a and 6b). The manuscript has been amended accordingly, lines 344 – 353)

Comment 8

Second, how are the diverse outcome metrics in "Figure 6 Data" aggregated into a single outcome

metric score?

Response 8

In preparing Figure 6, we did not aggregate the diverse health outcome metrics into a single outcome metric score. As there was a lack of variation in health outcome data across each of the programmes, within and between countries, whatever the mode of delivery, mix of personnel skill used, quantity of total personnel hours, quantity of education hours or quantity of participating patients, we chose instead to present this graphically based on one single illustrative health outcome variable, that of general diet. We hope that we have now clarified this decision sufficiently within the manuscript (lines 344-353), within the associated Supplemental File and in our response 7 above.

Comment 9

For the supporting data for Figure 6 Data, we need an explanation about the scale used by patients in reporting the PROs.

Response 9

Thank you, we have now inserted the precise scales used in the associated Supplemental File as a footnote to each of Tables 1-3 to ensure this is now clear to the reader.

Comment 10

The analysis claims that the outcomes are similar across programs, but does not address that the outcomes are also very similar to 0 (no impact from DMSE). Is the preponderance of positive signs used as the evidence that the DMSEs are weakly helpful, even though with the small sample sizes, virtually none of the changes is more than 1 SD away from 0.

Response 10

We agree with the reviewer, this is indeed the case. In many instances DSME was only weakly helpful or had no effect at all. This has now been highlighted within the manuscript (lines 350-353).

This may be due to the small sample size. We also believe it is due to the collection of self-reported health outcome data rather than the preferred clinical outcome data per participant before and after each DSME program. This has been highlighted in the Article Summary (lines 83-92) and in the Limitations section (lines 451-460) of the manuscript.

Comment 11

I don't understand the statement on lines 394-5 about statistically significant outcomes; only a very few of the reported means is more than 2x the standard deviation (what I assume SD stands for).

Response 11

Thank you for highlighting this to us. Within the EU comparative study, due to the small sample size of 366 participants, we did not analyze the health outcome data according to the different programmes but rather for all programmes in aggregate. Please see the new table now attached for the overall effectiveness for all programmes in aggregate within the Supplemental File Outcomes Framework and Outcomes Data, Table 3. We now realise that our reference on lines 394-395 to the 'statistically significant outcomes' was referring to this data and it was not clear to the reader why this statement was made. We have now referenced this piece of work regarding the health outcomes for all programs in aggregate within the revised manuscript on lines 404-412 (Peer et al., 2016). We hope that this now clarifies this statement.

Comment 12

I also don't see the data, in "Figure 6 Data" on the PROs of BMI, health related quality of life and

affective well-being (lines 396-7).

Response 12

Our apologies, this is now explained in response 11 above and these health outcomes can be found in this new Table 3: Overall Effectiveness of DSME programs in Aggregate within the Supplemental File Outcomes Framework and Outcomes Data.

Comment 13

Lines 430-433: It is unfortunate, but not fatal, that Belgium refused to supply confidential salary data. The Belgian study group could use data from public salary surveys of compensation of each type of employee. This would be approximately accurate, and good enough for the analysis in this paper, while not requiring that sensitive salary data be disclosed. I would encourage the Belgium group to access and apply public compensation survey data so they can rejoin the DMSE project team and paper.

Response 13

Yes it is indeed unfortunate that Belgium was unable to collect this data. We have reconnected with our partners in Belgium and confirm that the reason for this was related to data protection legislation (enacted 1992, subsequently amended 1998, 2003), together with the fact that there is no professional category of diabetes educator in Belgium.

Our Belgium colleagues have informed us that diabetes self-management education is provided by different professionals working in different types of organizations across Belgium, so there is likely to be a very large variation in salaries. A diabetes educator is not a job title in Belgium, so this category may comprise nurses, psychologists, social workers, and others working in different types of organisations (state hospitals, private hospitals, consultation centers, NGOs, self-employed status, ...). According to our Belgian colleagues there is no way to obtain a valid estimation of this cost without accessing data at personal level, and that is where the data privacy issue comes in.

Our colleagues have confirmed that while salary scales are available, they are meaningless as both the salary and the cost to the employer are dependent on many other variables including civil status, years of employment, fraction of employment, and type of organisation in which the employee works.

We have clarified this briefly within the manuscript, lines 445-449. We hope that this explains the absence of salary data from Belgium.

Thank you for your valuable comments and suggestions, which we have now reflected in the manuscript, Figures and Supplemental Files. We hope that we have thoroughly addressed your questions and suggestions and to have further improved this paper.

VERSION 3 – REVIEW

REVIEWER	Adriana Coppola Clinical Institute Beato Matteo, Vigevano, Italy
REVIEW RETURNED	27-Feb-2017

GENERAL COMMENTS	No additional comment
-----------------------